# Health service use among adults with cerebral palsy: a mixed methods systematic review protocol

Manjula Manikandan [ID],[1] Aisling Walsh,[1] Claire Kerr,[2] Michael Walsh,[3] Jennifer M Ryan [ID] [1,4]

¹Department of Public Health and Epidemiology, Royal College of Surgeons in Ireland, Dublin, Ireland
²School of Nursing and Midwifery, Queen's University Belfast, Belfast, UK
³Office of the Chief Clinical Officer, Health Service Executive, Dublin, Ireland
⁴College of Health, Medicine and Life Sciences, Brunel University London, Uxbridge, UK

**Correspondence to**
Dr Jennifer M Ryan;
jennifer.ryan@brunel.ac.uk

## ABSTRACT

**Introduction** Cerebral palsy (CP) is a neurodisability that primarily results in motor impairments and activity limitations, but is often associated with epilepsy and disturbances of sensation, perception, cognition, behaviour and speech. Most children with CP survive well into adulthood. Adults with CP experience increased risk of age-related chronic conditions such as arthritis, stroke, cardiorespiratory and mental health conditions in addition to the ongoing disabilities experienced from childhood. Therefore, adults with CP often require extensive health services. However, health service use among adults with CP has not been well documented. This mixed method review aims to identify, appraise and synthesise quantitative and qualitative literature examining health service use among adults with CP.

**Methods and analysis** The mixed method systematic review will be conducted in accordance with the Joanna Briggs Institute (JBI) methodology. A systematic search of MEDLINE (Ovid), CINAHL, Embase, PsycINFO and Cochrane Library from inception to March 2020 will be conducted. Quantitative observational studies, qualitative studies and mixed method studies examining health service use among adults with CP (≥18 years) will be included. Outcomes of interest are the proportion of adults using health services frequency of use and experiences of health services from the perspectives of adults with CP, caregivers and health service providers. Two reviewers will independently screen titles, abstracts and full-texts, extract data and assess the quality of included studies using JBI instruments. Where possible a pooled analysis and aggregation of findings will be performed for quantitative and qualitative data, respectively, and Grading of Recommendations Assessment, Development and Evaluation (GRADE)/GRADE-CERQual (Confidence in Evidence from Reviews of Qualitative research) employed. Quantitative and qualitative findings will be integrated using a triangulation approach at the synthesis stage. A narrative synthesis will be carried out where this is not possible.

**Ethics and dissemination** Ethical approval is not required for this review. The findings will be disseminated through a peer-reviewed journal and conferences.

**PROSPERO registration number** CRD42020155380.

### Strengths and limitations of this study

► This mixed method systematic review will include quantitative, qualitative and mixed methods studies in order to provide a comprehensive overview of the current literature relating to health services for adults with cerebral palsy.

► This review process is strengthened by utilisation of the widely used Andersen-Newman behavioural model of health service utilisation as an overarching framework to guide data extraction and analysis.

► Only studies that are published in English will be included in the review.

► Heterogeneity of eligible studies may limit the possibility of meta-analysis.

## BACKGROUND

Cerebral palsy (CP) is a "group of permanent disorders of movement and posture, causing activity limitation, that are linked to non-progressive disturbances of the developing fetal or infant brain".[1] The disturbance to the brain can occur before, during or up to approximately 2 years after birth from traumatic or non-traumatic brain injury.[2 3] With recent medical advances, CP can be diagnosed accurately before the age of 6 months.[4] People with CP present with different types of motor abnormality, namely spasticity, dyskinesia and ataxia.[1 5] In addition to movement disorders, people with CP may experience epilepsy or impairments of cognition, speech, hearing and vision.[1 5]

Traditionally, CP was considered a paediatric condition. The incidence rate of CP is approximately between 2.0 to 2.5 cases per 1000 live births worldwide.[6–9] However recently the birth prevalence has decreased in settings including Europe and Australia.[6–9] Many people with CP survive well into adulthood with a recent study reporting more than 80% of people with CP have a life expectancy, beyond 58 years.[10] People with CP continue to present with ongoing health issues from

**BMJ**

childhood, in addition to the issues related to ageing. For instance, adults with CP experience increased risk of arthritis, stroke, heart failure, ischaemic heart disease, chronic respiratory disease and mental health disorders compared with adults without CP.[11–14] Studies have also reported increased fatigue, incidence of falls, pain and a decline in mobility, as people with CP move into adulthood.[15–17] Therefore, adults with CP may require extensive health services.

The National Institute for Health and Care Excellence (NICE) guideline for CP in adults has recommended continuity of care for adults with CP, tailored to their needs.[18] This continuity of care is likely to avoid emergency hospital admissions.[18] However, studies of adolescents with CP have reported challenging experiences in navigating services during transition from child to adult services.[19 20] In addition, adolescents with CP reported anxiety about transfer from child to adult services as they perceived that services in adulthood were poorly co-ordinated and managed.[19–21] As the evidence describing the long-term health issues experienced by adults with CP has expanded in recent years, there is a growing need for research examining health service use among adults with CP.

'Health services' are defined as those 'services that are first and foremost undertaken to have direct effect on people's health. These extend from health promotion and disease prevention, through curative services, to long-term care and rehabilitation'.[22] Use of health services may be associated with various factors relating to the individual and their environment.[23 24] The previous Andersen-Newman behavioural models of health service utilisation (ANM) proposed family and individual as the unit of analysis for health service use.[24] The revised Andersen-Newman behavioural model of health service utilisation (ANM) provides a framework for considering health service use.[23] It describes environmental factors, population factors and outcomes that can be associated with health service use behaviour.[23] Environmental factors include those relating to the healthcare system (eg, national policies, resources and organisations involved) and external environment (eg, physical, political and economic factors).[23] Population factors include demographics (eg, age, gender, ethnicity, location and condition-specific characteristics such as functional mobility and type of motor abnormality).[23] Outcomes include perceived and evaluated health status and customer satisfaction.[23] Over the years, ANM has been adapted, critiqued and reviewed extensively.[25–29] However, the revised ANM (1995) includes environment as a key driver of the model impacting health service use behaviour.[23] It also emphasises the relationship between factors through feedback loops, which demonstrates the complexity involved in health service use behaviour.[23]

A preliminary search of PROSPERO, MEDLINE (Ovid), JoannaBriggs Institute (JBI) database and the Cochrane Database of Systematic Reviews indicated that no systematic reviews have been conducted examining health service use among adults with CP. Previous reviews have focussed on the epidemiology of CP in adults, quality of life, pain and effectiveness of interventions used by adults with CP.[16 30–32] There is a need for a theory informed review with broader perspectives on health service use among adults with CP, rather than focussing on the effectiveness of any specific intervention. Such a review would potentially inform health service used by adults with CP, direct research efforts and identify areas of challenges faced by adults with CP. Therefore, the aim of this mixed method systematic review is to identify, appraise and synthesise the available quantitative and qualitative literature examining health service use among adults with CP using the ANM as a framework.

## OBJECTIVES
The objectives are:
1. To determine the proportion and frequency of health service use by adults with CP.
2. To examine associations between environmental factors, population factors, outcomes and health service use.
3. To explore the experiences and perceptions of health service use for adults with CP, from the perspective of adults with CP, caregivers and health service providers.

## METHODS
This mixed method systematic review will be conducted in accordance with the JBI methodology guidelines.[33] The protocol was submitted for registration on the PROSPERO database of systematic reviews (registration number: CRD42020155 380). This review is written according to the Preferred Reporting Items for Systematic Reviewsand Meta-Analyses Protocols (PRISMA-P) checklist.[34] The review will be reported using Meta-analysis of observational studies inepidemiology (MOOSE) and Enhancing transparency in reporting the synthesis ofqualitative research (ENTREQ) reporting guidelines.[35 36]

### Study eligibility
#### Study design
The review will include quantitative, qualitative and mixed method studies. Quantitative studies will include observational studies (eg, cross-sectional, case-control and cohort studies). Qualitative studies will include studies of recognised qualitative approaches (eg, interviews, focus groups, observation, thematic analysis, content analysis, narrative analysis and framework analysis).[37] Mixed method studies will only be considered if quantitative or qualitative components can be clearly extracted.

Studies published in English language will be included. There will be no restriction on the geographical location of studies or date of publication.

### Participants
Studies of adults with CP (≥18 years) with all types of motor disorders and all levels of functional mobility will

be included for both quantitative and qualitative studies. Functional mobility for people with CP is widely classified using the Gross Motor Function Classification System.[38] For qualitative studies, articles including caregivers (also includes family members) and health service providers will also be included. For studies that include children and young adults (for example, those aged 15 to 25 years), data from people ≥18 years only will be included, if possible. If it is not possible to extract data for adults only, we will include data if the mean age of the sample is ≥18 years. Where studies include data on adults with CP as part of a larger sample of people with disabilities, we will extract data on adults with CP only, if possible.

### Outcome

Health services for this review will include services that have direct impact on the health of adults with CP.[22 39–42] This can include hospital admissions, emergency department visits, outpatient visits to medical and allied health professionals (eg, physicians, pharmacist, physiotherapists, occupational therapists, podiatrists, psychologists, orthotists, speech and language therapists). In addition, diagnostic or assistive device services (eg, radiologists, communication aids, mobility aids and wheelchair services) and support services (eg, supported living, personal assistance, residential, respite and domiciliary services) will also be included.[22 39–42]

For the quantitative component of this review, the proportion of adults using health services and the frequency of health service use will be the main outcomes. Where possible, we will report each type of health services used by adults with CP.

For the qualitative component, the main outcomes are experience and perceptions of using health services among adults with CP. We will also explore experiences and perceptions of health services for adults with CP from the perspectives of caregivers, and health service providers.

### Exclusion criteria

We will exclude reviews, dissertations, editorials, commentaries and conference abstracts. Studies of children and young people with CP (<18 years) will be excluded. Studies examining services that have an indirect effect on people's health only such as education, employment, housing and transport will be excluded.[22] Finally, randomised controlled studies on intervention effectiveness, pragmatic, open-label trials, case series and case reports will be excluded from this review.

### Search strategy

The electronic databases MEDLINE (Ovid), CINAHL, Embase, PsycINFO and Cochrane Library will be systematically searched from inception of each database to March 2020. We will conduct a pilot search in MEDLINE (Ovid) to inform the search strategy. A librarian will support development of the search strategy. Reference lists of included studies will be screened.

### Search terms

The search strategy will consist of fully exploded subject headings and free-text words relating to adults with CP and health services. The search strategy developed from MEDLINE (Ovid) will be tailored to each database. No filters on study design or settings will be applied at this stage. Search terms will be combined using Boolean operators. A sample search strategy for MEDLINE (Ovid) is provided in online supplementary appendix 1.

### Data management

A search log will be maintained to record the databases, keywords used and results of each search. The titles and abstracts of papers identified from the systematic search will be imported into reference management software (EndNote X8.2) and duplicates will be removed.

### Study selection

Two reviewers will screen titles and abstracts independently using Rayyan software.[43] Where studies meet the inclusion criteria or it is unclear if they are eligible, full-texts will be retrieved for further assessment. The two reviewers will screen full-text articles for inclusion independently. Conflicts will be resolved by discussion or third reviewer involvement. The reasons for exclusion of full-text studies will be recorded and the summary of the search results will be presented in a PRISMA flow diagram.[44]

### Data extraction

Two reviewers will extract data independently using the JBI data extraction tool for quantitative and qualitative studies accordingly. Any disagreements will be resolved initially through discussion and if required, consultation with a third reviewer. The study authors will be contacted to retrieve missing data necessary for meta-analysis or meta-aggregation.

### Data items

For quantitative studies and quantitative components of mixed method studies, extracted data will include study details, study design, country, funding sources, declaration of interests, participant characteristics, exposure and outcome of interest.[45] This will include proportion of people using each health service, frequency of use of each health service and variables relating to the environmental factors, population factors and outcomes according to the ANM. For qualitative studies and the qualitative components of mixed method studies, the extracted data will include study design, setting, country, methodology, funding sources, declaration of interests, participants, data analysis and themes related to experiences.[46] The qualitative data, such as quotes or themes of experiences and perceptions will be extracted from the results sections of the included studies.

### Assessment of methodological quality

Two reviewers will independently assess the methodological quality of included primary studies using standardised

critical appraisal instruments from the JBI.[33] For quantitative studies the appropriate JBI critical appraisal checklist for cohort, case-control and analytical cross-sectional studies will be used.[45] For qualitative studies the JBI qualitative appraisal instrument will be used.[46] Quality appraisal will be undertaken at study level and studies will not be excluded for data extraction or synthesis based on the methodological quality assessment, as this review seeks to capture all experiences of health service use by people with CP, and so necessitates inclusion of studies of lower methodological quality. Disagreements between the reviewers will be discussed and discrepancies resolved by a third reviewer. The results of the quality appraisal will be reported in a table and narrative form.

### Data synthesis
#### Quantitative studies
The pooled proportion of people using each type of health services and pooled mean visits to each type of health service will be estimated using a random effects model. We will standardise frequency to a specific time frame. Separate meta-analyses will be conducted for each service. We will use random-effects model due to the likely heterogeneity of the population and outcome variable.[47] We will assess statistical heterogeneity using the $I^2$ statistic and a $\chi^2$ test.[48] If studies compare health service use between people with and without CP, we will calculate an OR for dichotomous data and a standardised mean difference for continuous data. If data allow, we will conduct separate analysis for high-income countries and low- or middle-income countries as defined by the World Bank Countries classification to examine regional or income based variations in health service use among adults with CP.[49-51] Where data cannot be pooled, a narrative synthesis of the quantitative findings will be conducted.

#### Qualitative studies
Qualitative experiences from different perspectives will be pooled using a meta-aggregation approach.[52] This involves identifying the findings, categorising the findings based on similarity in meaning and grouping the categories into synthesised findings.[52] Where textual-pooling is not possible, the findings will be presented in narrative form.

#### Integration
The findings from quantitative and qualitative synthesis will be integrated using a convergent segregated approach, to preserve the integrity of quantitative and qualitative findings.[33] The results from quantitative and qualitative components will be triangulated and reported using the ANM as a framework. Triangulation will involve comparing or organising quantitative and qualitative findings into a line of argument via tabulation, groupings and visual representation to produce an overall configured analysis.[53] A narrative form of discussion will be considered, where this is not possible.

### Confidence in cumulative evidence
GRADE will be used to interpret the quantitative review findings and demonstrate the overall quality of the data.[54] GRADE-CERQual will be used to interpret the qualitative review findings in the context of methodological limitations, relevance, coherence and adequacy of data.[55] According to the JBI guidelines for mixed methods reviews, there are complexities involved in assessing confidence in findings from combined quantitative and qualitative evidence.[33] Therefore, the GRADE approach will only be used for separate qualitative and quantitative evidence analysis, but not for assessing the integrated findings

## ETHICS AND DISSEMINATION
Ethical approval is not required for this review as both quantitative and qualitative data used for analysis will be extracted from published studies. The review will be published in a peer-reviewed journal and presented at conferences for researchers, health professionals and people with CP. Findings will be disseminated to healthcare providers and shared with charity and voluntary agencies that support people with CP.

### Patient and public involvement
Adults with CP and professionals will be involved in interpretation and dissemination of the review findings.

## DISCUSSION
This mixed methods review will provide greater insights of the available literature on the health services used by adults with CP and related experiences of health services from the perspectives of adults with CP, their caregivers and health service providers. The majority of people with CP will survive to at least 60 years of age.[10] Many adults with CP experience secondary conditions with age that require intervention and support from clinicians working across multiple health and social care disciplines.[12] Previous research has demonstrated that young people with CP experience challenges in accessing health services, as they move from child to adult health services.[19 21 56-58] Therefore, a review of health service use among adults with CP is required in order to direct research efforts and inform service provision.

A theory-informed review using the revised ANM will guide examination of factors associated with health service use among adults with CP.[23] This model describes factors by environment, population and outcomes influencing health service use behaviour with feedback loops showing the complexities involved.[23] However, the ANM provides only a cross-sectional view on health service use and does not include change or life course as a factor.[29] This incomplete view of a phenomenon is common among most theoretical frameworks used in the literature.[59] The life course perspective provides a complete understanding of past versus present service use and

the changes seen over time.[29] Where possible a narrative synthesis will be done for life course or change over time in service use.

A limitation of this review is the exclusion of studies that are not peer-reviewed or published in languages other than English, potentially resulting in less generalisable findings from other languages. Finally, it may not be possible to conduct a meta-analysis due to a lack of / and/or heterogeneity of data.

**Acknowledgements** The authors acknowledge the support from the RCSI librarian (Ms Grainne McCabe) for supporting the development of search terms with the reviewers.

**Contributors** All authors have agreed on the final version and meet at least two of the following criteria (recommended by ICMJE www.icmje.org/recommendations): Substantial contributions to the conception or design of the work; or the acquisition, analysis or interpretation of data for the work (MM, AW, CK, MW and JR). Drafting the work or revising it critically for important intellectual content (MM and JR). Final approval of the version to be published (MM, AW, CK, MW and JR). Agreement to be accountable for all aspects of the work in ensuring that questions related to the accuracy or integrity of any part of the work are appropriately investigated and resolved (MM and JR).

**Funding** This work was conducted as part of the SPHeRE Programme under Grant No. SPHeRE/2013/1. This review is funded by the Royal College of Surgeons in Ireland (RCSI) through the StAR programme. The funders had no role in developing the protocol.

**Competing interests** None declared.

**Patient and public involvement** Patients and/or the public were involved in the design, or conduct, or reporting, or dissemination plans of this research. Refer to the Methods section for further details.

**Patient consent for publication** Not required.

**Provenance and peer review** Not commissioned; externally peer reviewed.

**ORCID iDs**
Manjula Manikandan http://orcid.org/0000-0003-2631-8482
Jennifer M Ryan http://orcid.org/0000-0003-3768-2132

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
