## [Reviewer comments · BMJ Open]

ARTICLE DETAILS

TITLE (PROVISIONAL)	HEALTH SERVICE USE AMONG ADULTS WITH CEREBRAL PALSY: A MIXED METHODS SYSTEMATIC REVIEW PROTOCOL.
AUTHORS	Manikandan, Manjula; Walsh, Aisling; Kerr, C; Walsh, Michael; Ryan, Jennifer

VERSION 1 – REVIEW

REVIEWER	Emily Shepherd The University of Adelaide, Australia
REVIEW RETURNED	21-Jan-2020

GENERAL COMMENTS	Thank you for the opportunity to review this protocol, which is well written. The proposed review will address an important question. I have some minor comments/suggestions below for the authors to consider: * Consider revising the title to 'Health service use among adults with cerebral palsy: a mixed methods systematic review protocol'* In the background, consider updating the statement regarding cerebral palsy diagnosis to include reference to the new international diagnosis guidelines (Novak et al. 2017), and consider referencing a systematic review of associated impairments (Novak et al. 2012). Recent evidence suggests that the birth prevalence of cerebral palsy is in fact declining for the first time - consider revising your statement stating it is 'increasing' (e.g. Galea et al. 2019).* Inclusion criteria: will you consider any publication year? Consider changing 'caregivers' to capture family members who may not be labelled as such; and 'health professionals' to include health service providers* Your search date was November 2019 - most journals will require a search to be within 6 months of publication, so you may require a 'top up' search* Data items: you may like to include funding source/s and declarations of interest in the items extracted* Data synthesis: can you justify the use of a fixed-effects model?* Discussion: is the first paragraph necessary? (repetitive)* Reference list requires some editing for consistency (e.g. journal names, capitalisation of article titles)* Re: PRISMA-P statement - re-consider whether you have covered the last item, #17* Re: Search strategy - have you piloted this? It looks like it will be very broad - with the use of terms such as therapy/diagnosis/health care/hospital AND cerebral palsy AND adult, likely to attract many results. You may wish to reconsider this? Best wishes
--

REVIEWER	Joseph Nguemo Ryerson University, Canada
REVIEW RETURNED	09-Feb-2020

GENERAL COMMENTS	A MIXED METHODS SYSTEMATIC REVIEW PROTOCOL OF HEALTH SERVICE USE AMONG ADULTS WITH CEREBRAL PALSY This study attempts to address one important issue about health care use among individuals with CP. Overall, I would agree that this is a great study given the complexity of the disease and challenges face this by this group of individuals in their day-to-day life. Over all the paper is well writing and flows very well. Although I lean more towards acceptance of this article, it is important for the authors to review my comments., particular regarding the use and application of the JBI systematic method. You will find below some comments regarding the article Abstract Line: 13: Can you provide some examples of secondary conditions experienced by CP adults? I also think that the sentence “As adults..... the ongoing health issues from childhood” does not sound good. Maybe you have to rephrase it differently. It sounds to me that they only experience secondary conditions at adulthood? This means that there are no secondary conditions to experience in childhood? Line 15-17: I am not sure what do you mean by “health service useremains unclear”. does it mean health service use is not been well documented? Line 36: I think the sentence “...the experience and perceptions of health services for” is incomplete. Line 52-54: you can just say that “the results will be disseminated through relevant peer review journals and conferences.” I don’t think important to list the audiences. This is up to you. STRENGTHS & LIMITATIONS OF THIS STUDY Line 16: Is the Andersen-Newman behavioral model is a strength or limitation? Indicate why it a strength or limitation Background Line 41-52: the sentence “as adults.....related to ageing”. It looks like the sentence is incomplete Line 20. What is the limitation of including only studies published in English? Over all, I don’t see the rationale of conduction this review. Why this review is important? What will this review add in the current body of knowledge? Objective Please rephrase the objectives. But this is up to you.  e.g “to determine the proportion and frequency of health service use by adults with CP” e.g To examine associations environmental factors, population factors, outcomes and health care use . I leave this up to you
--

3. e.g to explore the experiences and perceptions of adults with CP and service providers about health use by adults with CP

Method

Line 42. Provide proper and complete reference for the mixed method systematic review according to JBI :
Lizarondo L, Stern C, Carrier J, Godfrey C, Rieger K, Salmond S, Apostolo J, Kirkpatrick P, Loveday H. Chapter 8: Mixed methods systematic reviews. In: Aromataris E, Munn Z (Editors). Joanna Briggs Institute Reviewer's Manual. The Joanna Briggs Institute, 2017. Available from <https://reviewersmanual.joannabriggs.org/>

Study eligibility

Study design

Line 6. are authors considering include observation studies such case series or case reports?

Line 27: The description of the population of interest is not clear. Who will be included in quantitative studies.

Also, it is not clear if adults with adults CP will be included in qualitative studies. But, based on objective 3, I can assume that they will be included but the authors have to said it.

Exclusion

Are you excluding pragmatic, open-label trials?

Search strategy

Line 42: Any reason for limiting the search to Nov 2019? Why not from inception to date?

Search terms

Line 6: I would suggest revising your search strategy and include more search terms such as: podiatrists, psychologists, orthotists, audilogists?

What about audilogists? It looks like people with CP might experience hearing issues?

P9 Line 16: are you using any software for screening?

Also, clarify if you will record the reasons for exclusion of full-text studies that do not meet the inclusion criteria in the study selection section in the PRISMA graph (I strongly recommend this)

Data items

Line 51. You said a data extraction form will be developed and in the abstract, you said that "Two reviewers will independently screen titles, abstracts and full texts, extract data, and assess the quality of included studies using instruments from the JBI". Please clarify whether you will use the JBI data extraction form

Quality assessment

Line 25; Please review JBI method for quality assessment for mixed methods. The assessment of quantitative and qualitative papers requires 2 different methods (chap 3 and chap 2 of the manual).

Are you excluding some papers based on the quality assessment?

Are all studies, regardless of the results of their methodological quality, will undergo data extraction and synthesis?

	Data synthesis Line 42: how are you going to express the effect sizes? (as odd ratio or weighted? (or standardized?) Please clarify Line 44: any rationale for using random effects over fixed-effects? Please clarify Are you conducting any sensitivity analysis? Please clarify Are you generating any funnel plot to assess publication bias? Please clarify Integration Line 21: I think reference 30 is out to data. (please check the following ref: Chapter 8: Mixed methods systematic reviews. In: Aromataris E and Munn Z, eds. Joanna Briggs Institute Reviewer's Manual, The Joanna Briggs Institute)
--	---

REVIEWER	Olga Petrovskaya University of Alberta
REVIEW RETURNED	17-Feb-2020

GENERAL COMMENTS	Thank you for the opportunity to review this manuscript.  1) Consider adding the following keywords: protocol; systematic review 2) Consider adding to the Abstract that the phenomenon of interest will be addressed from the perspectives of caregivers and health professionals (not just adults with CP). This is made clear on p. 5, but not in the Abstract. 3) Provide PROSPERO registration number. 4) On p. 5 you indicate that a preliminary search was conducted to locate existing systematic reviews (SRs) on the topic. Consider searching the JBI database as an additional source. 5) What is the rationale for not restricting your searches to specific geographic location(s) and time period(s)? Clearly, health and social service provision in the Western countries has shifted significantly over the last 2 decades (neoliberal policies, self-care ideology with increased roles of patients and families, budget constrains, quality assurance campaigns etc). In turn, outside of the Western context, these services probably have always been and continue to be quite different from those in the Anglophone westernized healthcare systems. It is not clear how the summary and integration of findings from these various contexts and time periods can be meaningful? Consider setting restrictions? 6) Based on the same observation (point 5 above) that the context of health and social services has changed tremendously over the last 2 decades (also including technological advances eg new rehab devices, patients' access to their health information via web portals, providers' EMRs, care pathways), it appears that the two important sources chosen as a theoretical framework for this SR might not be appropriate (ie, outdated): Andersen-Newman dated 1973 and Leutz dated 1999. Were there any updates to these papers, esp. ANM? If not, perhaps consider another framework or elaborate on the limitations of ANM and consider how these limitations will be mitigated. 7) and 8) These two comments relate to items #14 and #17 from the Reporting checklist (PRISMA-P). It appears that the authors are conflating quality assessment of the primary studies retrieved during searches and the assessment of confidence in cumulative evidence.
---

	While the Quality Assessment is described on p. 9, the order of this step in the overall review methodology and the proposed process should be revisited in light of the JBI recommendations. Specifically, I am quoting from the following source: https://wiki.joannabriggs.org/display/MANUAL/8.4.2+++MMSR+questions+that+take+a+CONVERGENT+SEGREGATED+approach+to+synthesis+and+integration Assessment of methodological quality "Studies that are eligible for inclusion in the review must be assessed for methodological quality. The decision as to whether or not to include a study can be made based on meeting a pre-determined proportion of all criteria, or on certain criteria being met. It is also possible to weight certain criteria differently. Decisions about a scoring system or any cut-off for exclusion should be made in advance and agreed upon by all reviewers before critical appraisal commences. All included studies need to be critically appraised using the standard JBI critical appraisal instruments" Please elaborate on how are you planning to use the quality appraisal instrument and how you will handle low quality primary studies (ie, those with a low score). 8) Finally, in terms of the assessment of confidence in cumulative evidence (#17 in the Reporting guidelines). This is a separate step from the Quality Appraisal above. If the QA is done BEFORE the data extraction and analysis/synthesis in order to determine what studies should be included, Assessment of Confidence in Evidence refers to the evaluation of YOUR findings and is the concluding step in the SR methodology. Although the JBI does not recommend using GRADE, this only applies to manuscripts considered for publication in JBI journals. In contrast, BMJ follows PRISMA-P guidelines and thus requires the assessment of confidence in cumulative evidence. Therefore, please include this step and elaborate on how you will apply GRADE for quantitative evidence and CERQual for qualitative evidence. Thank you!
--	--

VERSION 1 – AUTHOR RESPONSE

Reviewer’s comments and Author’s responses

S.N o.	Reviewer comments	Author’s responses
Reviewer 1: Emily Shepherd		
1.	Consider revising the title to 'Health service use among adults with cerebral palsy: a mixed methods systematic review protocol'	Title updated accordingly.

2.	In the background, consider updating the statement regarding cerebral palsy diagnosis to include reference to the new international diagnosis guidelines (Novak et al. 2017), and consider referencing a systematic review of associated impairments (Novak et al. 2012). Recent evidence suggests that the birth prevalence of cerebral palsy is in fact declining for the first time - consider revising your statement stating it is 'increasing' (e.g. Galea et al. 2019).	Added Reference 2 and 3 on CP brain disturbance from traumatic or non-traumatic injury. Novak et al 2017 (ref 4) added to the first paragraph in the background on CP diagnosis guidelines. Novak et al 2012 added to the first paragraph in the background (reference no. 5) Ref 9 and 10 on decreasing prevalence in Australia and Europe updated to the second paragraph in the background.
3.	Inclusion criteria: will you consider any publication year? Consider changing 'caregivers' to capture family members who may not be labelled as such; and 'health professionals' to include health service providers	Updated under study design- no restriction on date of publication. Caregivers can also include family members (updated in Inclusion criteria-Participants). Changed health professionals to health service providers.
4.	Your search date was November 2019 - most journals will require a search to be within 6 months of publication, so you may require a 'top up' search	Updated under search strategy- Inception to March 2020, email alerts are set up in all databases to make sure that the review is current.
5.	Data items: you may like to include funding source/s and declarations of interest in the items extracted	Updated funding sources and declarations of interest under data items accordingly.
6.	Data synthesis: can you justify the use of a fixed-effects model?	Random effects model justified over fixed effect (ref 49) as follows: “We will use random-effects model due to the likely heterogeneity of the population and outcome variable”.

7.	Discussion: is the first paragraph necessary? (repetitive)	Removed and updated accordingly.
8.	Reference list requires some editing for consistency (e.g. journal names, capitalisation of article titles)	Updated reference from Vancouver to BMJ template in End note with Italics for Journal names. Manuscript reference is updated accordingly with superscript vancouver (BMJ style). The consistency of journal names checked and updated accordingly.
9.	Re: PRISMA-P statement - reconsider whether you have covered the last item, #17	GRADE and GRADE CerQual has been updated to the manuscript accordingly.
10.	Search strategy - have you piloted this? It looks like it will be very broad - with the use of terms such as therapy/diagnosis/health care/hospital AND cerebral palsy AND adult, likely to attract many results. You may wish to reconsider this?	Yes, the search strategy was piloted in Medline (Ovid) and it resulted in 5700 articles initially. From the first 100 articles, we found 3 articles relevant to the aim of this study. We agree that the search strategy is broad so, that we don't miss any articles relevant to health service use among adults with CP.
Reviewer 2: Joseph Nguemo		
Abstract:		
1.	Line: 13: Can you provide some examples of secondary conditions experienced by CP adults? I also think that the sentence "As adults..... the ongoing health issues from childhood" does not sound good. Maybe you have to rephrase it differently. It sounds to me that they only experience secondary conditions at adulthood? This means that there	Example added in the abstract accordingly. The sentence is paraphrased as follows: "Adults with CP experience increased risk of age related chronic conditions such as arthritis, stroke, cardio respiratory and mental health conditions in addition to the ongoing childhood disability".

	are no secondary conditions to experience in childhood?	
2.	Line 15-17: I am not sure what do you mean by “health service useremains unclear”. does it mean health service use is not been well documented?	Updated to “health service use has not been well documented”.
3.	Line 36: I think the sentence “...the experience and perceptions of health services for” is incomplete.	The sentence is updated as follows: “experiences of health services use from the perspectives of adults with CP, caregivers and health service providers”.
4.	Line 52-54: you can just say that “the results will be disseminated through relevant peer review journals and conferences.” I don’t think important to list the audiences. This is up to you.	Updated to “peer review journals and conferences” only.
Strengths & Limitations of this study:		
5.	Line 16: Is the Andersen-Newman behavioral model is a strength or limitation? Indicate why it a strength or limitation	Andersen-Newman behaviour model (1995) is used as a strength in guiding the review process. It is widely used model in the health service use literature since 1973 and revised extensively in 1995 model. The broad & revised ANM model (1995) will be used in this review. The manuscript has been updated accordingly on Page 3 strengths & limitations and Page 5 background sections.
Background:		
6.	Line 41-52: the sentence “as adults.....related to ageing”. It looks like the sentence is incomplete Line 20. What is the limitation of including only studies published in English? Over all, I don’t see the rationale of conduction this review. Why	Removed ‘as adults’ to make it a complete sentence. Limitation of using English language studies are less generalisable findings. It has been updated in the discussion section.

	this review is important? What will this review add in the current body of knowledge?	The rationale for conducting the review has been updated in the background on Page 5 and 6 accordingly.
Objective		
7.	Please rephrase the objectives. But this is up to you. 1. e.g “to determine the proportion and frequency of health service use by adults with CP” 2. e.g To examine associations environmental factors, population factors, outcomes and health care use . I leave this up to you 3. e.g to explore the experiences and perceptions of adults with CP and service providers about health use by adults with CP	Updated Objective 1 and 2. Objective 3 updated to ‘health service use’.
Method		
8.	Line 42. Provide proper and complete reference for the mixed method systematic review according to JBI : Lizarondo L, Stern C, Carrier J, Godfrey C, Rieger K, Salmond S, Apostolo J, Kirkpatrick P, Loveday H. Chapter 8: Mixed methods systematic reviews. In: Aromataris E, Munn Z (Editors). Joanna Briggs Institute Reviewer's Manual. The Joanna Briggs Institute, 2017. Available	Reference updated accordingly.

	from https://reviewersmanual.joannabriggs.org/	
Study eligibility		
9.	Study design Line 6. are authors considering include observation studies such case series or case reports?	Excluded case series and case reports, updated exclusion criteria accordingly.
10.	Line 27: The description of the population of interest is not clear. Who will be included in quantitative studies.	Updated under participant section accordingly as follows: “for both quantitative and qualitative studies”
11.	Also, it is not clear if adults with adults CP will be included in qualitative studies. But, based on objective 3, I can assume that they will be included but the authors have to said it.	Updated under participant section accordingly.
Exclusion		
12.	Are you excluding pragmatic, open-label trials?	Yes, excluded and updated exclusion criteria accordingly.
Search strategy		
13.	Line 42: Any reason for limiting the search to Nov 2019? Why not from inception to date?	Updated the search from inception to March 2020.
14.	Search terms : Line 6: I would suggest revising your search strategy and include more search terms such as: podiatrists,	Because our search terms are broad on all health services, it will incorporate podiatrists, orthotists and audiologists, similar to Physiotherapist, Occupational therapist, Nurse, Physicians, Consultants, Speech & language therapist, etc,. Our focus was more on health services and not health personnel/occupation as such.

	psychologists, orthotists, audiologists? What about audiologists? It looks like people with CP might experience hearing issues?	The fully exploded 'Health services' includes  • Mental health services that captures community mental health services and counselling services. The fully exploded Mesh term 'Analytical, Diagnostic and Therapeutic Techniques and Equipment' includes  • prosthesis fitting and • prosthesis and implants (bioprosthesis, absorbable implants) • Joint prosthesis • Self-help devices (communications aids for disabled and wheelchairs) The fully exploded term 'Analytical, Diagnostic and Therapeutic Techniques and Equipment' includes  • Rehabilitation (correction of hearing impairment, manual communication) • Prosthesis and implants (cochlear implants) • sensory aids (hearing aids) • rehabilitation of speech and language disorders therapies. Thus, the studies that examine services provided by these professions will be identified using the current search strategy.
15.	P9 Line 16: are you using any software for screening? Also, clarify if you will record the reasons for exclusion of full-text studies that do not meet the inclusion criteria in the study selection section in the PRISMA graph (I strongly recommend this)	Rayyan will be used to screen title and abstracts: the manuscript has been updated to reflect this. Clarification provided in the section entitled 'study selection' as follows: "The reasons for exclusion of full text studies will be recorded and the summary of the search results will be presented in a Preferred Reporting Items for Systematic Reviews and Meta-analyses (PRISMA) flow diagram"
Data items		
16.	Line 51. You said a data extraction form will be developed	Yes, we will use JBI data extraction tool for quantitative studies (case-control, cohort and Analytical cross-

	and in the abstract, you said that “Two reviewers will independently screen titles, abstracts and full texts, extract data, and assess the quality of included studies using instruments from the JBI”. Please clarify whether you will use the JBI data extraction form	sectional studies- Chapter 7.3.6.4-Data extraction) and for qualitative studies (JBI manual Chapter 2-Appendix 2.3). The manuscript has been updated under data extraction and data items, accordingly.
Quality assessment		
17.	Line 25; Please review JBI method for quality assessment for mixed methods. The assessment of quantitative and qualitative papers requires 2 different methods (chap 3 and chap 2 of the manual).	Chapter 2 & 3 of the manual (qual & quant) have been reviewed: The qualitative appraisal JBI tool is suitable (chapter 2). The quantitative appraisal tool (chapter 3) is for RCT and non-randomised experimental studies (looking at intervention), (chapter 3 -appendix 3.1-3.4). However, the aim of our review is not analysing the effectiveness of interventions so this tool is not appropriate in this instance. https://wiki.joannabriggs.org/pages/viewpage.action?pageId=9273720 Instead, JBI Chapter 7 (systematic reviews of etiology and risks) includes an appraisal tool for cohort studies (Appendix 7.1), case control studies (Appendix 7.2) and analytical cross-sectional studies (Appendix 7.5) that will be used accordingly. Manuscript referenced with chapter 2 and 7 and updated as follows: “For quantitative studies appropriate JBI- critical appraisal checklist for cohort, case-control and analytical cross-sectional studies will be used. Similarly, for qualitative studies the JBI- qualitative appraisal instrument will be used”

	Are you excluding some papers based on the quality assessment? Are all studies, regardless of the results of their methodological quality, will undergo data extraction and synthesis?	No, studies will not be excluded based on the quality assessment. The manuscript has been updated accordingly. Yes, all studies, regardless of methodological quality, will undergo data extraction and synthesis. This has been reflected in the updated manuscript under quality assessment as follows: “Studies will not be excluded for data extraction or synthesis based on the quality assessment”.
Data synthesis		
18.	Line 42: how are you going to express the effect sizes? (as odd ratio or weighted? (or standardized?)) Please clarify	Estimates of effect are not anticipated in this review as the review is not examining the effect of any interventions or determining the effect of health service use. Instead, we will report the pooled proportion of adults using each health service and the pooled mean number of visits. If studies compare health service use between people with and without CP, we will calculate an odds ratio for dichotomous data and a standardised mean difference for continuous data.
19.	Line 44: any rationale for using random effects over fixed-effects? Please clarify Are you conducting any sensitivity analysis? Please clarify	Random effects model justified over fixed effect (ref 49) as follows in the manuscript: “We will use random-effects model due to the likely heterogeneity of the population and outcome variables. Sensitivity analysis will be not be done.

20.	Are you generating any funnel plot to assess publication bias? Please clarify	Funnel plots of effect sizes from individual studies will not be carried out to assess publication bias as this review will not evaluate the effectiveness of specific interventions.
Integration		
21.	Line 21: I think reference 30 is out to data. (please check the following ref: Chapter 8: Mixed methods systematic reviews. In: Aromataris E and Munn Z, eds. Joanna Briggs Institute Reviewer's Manual, The Joanna Briggs Institute)	The manuscript has been updated with the suggested reference (33).
Reviewer 3: Olga Petrovskaya		
1.	Consider adding the following keywords: protocol; systematic review	Added two new keywords accordingly.
2.	Consider adding to the Abstract that the phenomenon of interest will be addressed from the perspectives of caregivers and health professionals (not just adults with CP). This is made clear on p. 5, but not in the Abstract	Abstract updated accordingly.
3.	Provide PROSPERO registration number.	The protocol is under review at PROSPERO [receipt number: 155380].
4.	On p. 5 you indicate that a preliminary search was conducted to locate existing systematic reviews (SRs) on the topic. Consider searching the JBI database as an additional source.	JBI database searched and the manuscript has been updated accordingly.
5.	What is the rationale for not restricting your searches to specific geographic location(s) and time period(s)?	Geographic location can itself be a factor associated with health service use. Where possible those characteristics will be reviewed and synthesised. The JBI data extraction tool will allow context/setting to be analysed, and this will be included in the results and

	Clearly, health and social service provision in the Western countries has shifted significantly over the last 2 decades (neoliberal policies, self-care ideology with increased roles of patients and families, budget constrains, quality assurance campaigns etc). In turn, outside of the Western context, these services probably have always been and continue to be quite different from those in the Anglophone westernized healthcare systems. It is not clear how the summary and integration of findings from these various contexts and time periods can be meaningful? Consider setting restrictions?	discussion sections accordingly. Also, the ANM will assist with discussing the contextual factors. We anticipate that using ANM as a theoretical framework will allow us to consider contexts (high/middle/ low income) and time periods, where possible. If this is not possible, a narrative synthesis of these will be undertaken.
6.	Based on the same observation (point 5 above) that the context of health and social services has changed tremendously over the last 2 decades (also including technological advances eg new rehab devices, patients' access to their health information via web portals, providers' EMRs, care pathways), it appears that the two important sources chosen as a theoretical framework for this SR might not be appropriate (ie, outdated): Andersen-Newman dated 1973	We anticipate that using ANM as a theoretical framework will allow us to consider contexts (high/middle/ low income) and time periods, where possible. If this is not possible, a narrative synthesis of these will be undertaken. The Andersen-Newman behaviour model (1995) will be used to guide the review process. It is a widely used model in the health service use literature since 1975

	and Leutz dated 1999. Were there any updates to these papers, esp. ANM? If not, perhaps consider another framework or elaborate on the limitations of ANM and consider how these limitations will be mitigated.	and was revised extensively in 1995. The broader and revised ANM (1995) will be used in this review. The manuscript has been updated accordingly in strengths & limitations (Page 3) and background section (Page 5) and discussion (2nd paragraph) with justifications & limitations. Leutz 1999 is reference for the definition of health & social services and not theoretical framework.
7.	7) and 8) These two comments relate to items #14 and #17 from the Reporting checklist (PRISMA-P). It appears that the authors are conflating quality assessment of the primary studies retrieved during searches and the assessment of confidence in cumulative evidence. While the Quality Assessment is described on p. 9, the order of this step in the overall review methodology and the proposed process should be revisited in light of the JBI recommendations. Specifically, I am quoting from the following source: https://bit.ly/3bKA3hU Assessment of methodological quality "Studies that are eligible for inclusion in the review must be assessed for methodological quality. The decision as to whether or not to include a study can be made based on meeting a pre-determined proportion of all criteria, or on certain criteria being met. It is also possible to weight certain criteria differently. Decisions about a scoring system	The quality appraisal will be undertaken at study level rather than outcome level. The studies will not be excluded based on quality assessment as this might end in small number of studies being eligible to review. This review is novel, complex and exploratory in many ways and we would like to capture all data. The manuscript is updated accordingly under quality assessment.

	or any cut-off for exclusion should be made in advance and agreed upon by all reviewers before critical appraisal commences. All included studies need to be critically appraised using the standard JBI critical appraisal instruments" Please elaborate on how are you planning to use the quality appraisal instrument and how you will handle low quality primary studies (ie, those with a low score).	Chapter 2 & 3 of the manual (qual & quant) have been reviewed: The qualitative appraisal JBI tool is suitable (chapter 2). The quantitative appraisal tool (chapter 3) is for RCT and non-randomised experimental studies (looking at intervention), (chapter 3 -appendix 3.1-3.4). However, the aim of our review is not analysing the effectiveness of interventions so this tool is not appropriate in this instance. https://wiki.joannabriggs.org/pages/viewpage.action?pageId=9273720 Instead, JBI Chapter 7 (systematic reviews of etiology and risks) includes an appraisal tool for cohort studies (Appendix 7.1), case control studies (Appendix 7.2) and analytical cross-sectional studies (Appendix 7.5) that will be used accordingly. Manuscript referenced with chapter 2 and 7 and updated as follows: "For quantitative studies appropriate JBI- critical appraisal checklist for cohort, case-control and analytical cross-sectional studies will be used. Similarly, for qualitative studies the JBI- qualitative appraisal instrument will be used"
--	--	---

		The difference between high and low quality primary studies will be narratively described. Low quality primary studies will not be excluded by the reviewer. This will be included in the review and discussed accordingly in the synthesis section.
8.	Finally, in terms of the assessment of confidence in cumulative evidence (#17 in the Reporting guidelines). This is a separate step from the Quality Appraisal above. If the QA is done BEFORE the data extraction and analysis/synthesis in order to determine what studies should be included, Assessment of Confidence in Evidence refers to the evaluation of YOUR findings and is the concluding step in the SR methodology. Although the JBI does not recommend using GRADE, this only applies to manuscripts considered for publication in JBI journals. In contrast, BMJ follows PRISMA-P guidelines and thus requires the assessment of confidence in cumulative evidence. Therefore, please include this step and elaborate on how you will apply GRADE for quantitative evidence and CERQual for qualitative evidence.	The studies will not be excluded based on the quality assessment. So, the data extraction and quality appraisal will be done for all the included studies. There is no GRADE approach for assessing confidence in findings developed from combined quantitative and qualitative evidence. Therefore, we will not be able to apply a certainty rating to the integrated analysis. GRADE and GRADE CERQual will be used for the assessment of the evidence for the quantitative and qualitative analysis, but not for integrated analysis. The manuscript has been updated and referenced accordingly under quality assessment section.

VERSION 2 – REVIEW

REVIEWER	Emily Shepherd The University of Adelaide, Australia
REVIEW RETURNED	06-Apr-2020

GENERAL COMMENTS	Thank you for the revision, which I believe nicely addresses all 3 reviewer comments, and certainly strengthens the protocol. Minor edits  - Consider adding evidence strength (GRADE) assessment to abstract - Background: Birth prevalence has decreased in other settings too - so suggest "in settings including Europe and Australia..." - Methods: like another reviewer, I do have remaining concerns regarding the pooling of data from such wide ranging settings - I am not sure such pooled results would be meaningful. Would the authors consider prespecifying subgroup analyses (or indeed separate analyses) based on high and low/middle income countries? Best wishes
---

REVIEWER	Olga Petrovskaya University of Alberta
REVIEW RETURNED	04-Apr-2020

GENERAL COMMENTS	Thank you for revising the manuscript. Please consider the following minor additional revision: P 11 Paragraph starting on the line 19  1. Make explicit that this paragraph refers to assessing the quality of findings produced in YOUR review (in contrast to the preceding paragraph focused on assessing the quality of primary studies included in the review). 2. During what stage of data synthesis (described on pp. 10-11) will you apply GRADE and GRADE-Cerqual? 3. Consider moving this paragraph toward the end of the protocol for consistency with the logical progression of the review process, as illustrated by the item # 17 "Confidence in cumulative evidence" in the checklist. Thank you!
---

VERSION 2 – AUTHOR RESPONSE

S.N o.	Reviewer comments	Author's responses
Reviewer 1: Emily Shepherd		
1.	Please state any competing interests or state 'None declared': None	Competing interest updated to 'None declared' in Page number 14.
2.	Consider adding evidence strength (GRADE) assessment to abstract	Abstract has been updated with GRADE and GRADE-CERQual accordingly.

3.	Background: Birth prevalence has decreased in other settings too - so suggest "in settings including Europe and Australia..."	Background has been updated accordingly in page number 4.
4.	Methods: like another reviewer, I do have remaining concerns regarding the pooling of data from such wide ranging settings - I am not sure such pooled results would be meaningful. Would the authors consider prespecifying subgroup analyses (or indeed separate analyses) based on high and low/middle income countries?	We will conduct separate analysis for high and low/middle income countries, where possible. Additionally, we believe that we will be able to identify geographic differences via our data extraction and use of the ANM framework. The manuscript has been updated accordingly.
Reviewer 3: Olga Petrovskaya		
1.	P 11 Paragraph starting on the line 19 Make explicit that this paragraph refers to assessing the quality of findings produced in YOUR review (in contrast to the preceding paragraph focused on assessing the quality of primary studies included in the review).	The Page 11, line 19 paragraph is the quantitative data synthesis section and not assessing quality of findings. The quality appraisal title is updated as 'Assessment of methodological quality' as suggested, and Confidence in cumulative evidence section added, after data synthesis section, which includes GRADE i.e quality of findings produced in the review.
2.	During what stage of data synthesis (described on pp. 10-11) will you apply GRADE and GRADE-Cerqual?	GRADE will be done after data extraction, quality appraisal and synthesis of quantitative and qualitative findings, but before integration of both findings.
3.	Consider moving this paragraph toward the end of the protocol for consistency with the logical progression of the review process, as illustrated by the item # 17 "Confidence in cumulative evidence" in the checklist.	The GRADE paragraph moved under Confidence in cumulative evidence section as suggested in page 12.